# How Language Models Prioritize Contextual Grammatical Cues?

**Hamidreza Amirzadeh** [1]  **Afra Alishahi** [2]  **Hosein Mohebbi** [2]

[1] Sharif University of Technology, Iran  [2] Tilburg University, the Netherlands

hamid.amirzadeh78@sharif.edu

{a.alishahi, h.mohebbi}@tilburguniversity.edu

## Abstract

Transformer-based language models have shown an excellent ability to effectively capture and utilize contextual information. Although various analysis techniques have been used to quantify and trace the contribution of single contextual cues to a target task such as subject-verb agreement or coreference resolution, scenarios in which multiple relevant cues are available in the context remain underexplored. In this paper, we investigate how language models handle gender agreement when multiple gender cue words are present, each capable of independently disambiguating a target gender pronoun. We analyze two widely used Transformer-based models: BERT, an encoder-based, and GPT-2, a decoder-based model. Our analysis employs two complementary approaches: context mixing analysis, which tracks information flow within the model, and a variant of activation patching, which measures the impact of cues on the model's prediction. We find that BERT tends to prioritize the first cue in the context to form both the target word representations and the model's prediction, while GPT-2 relies more on the final cue. Our findings reveal striking differences in how encoder-based and decoder-based models prioritize and use contextual information for their predictions.

## 1 Introduction

Pre-training language models on large data using the Transformer (Vaswani et al., 2017) architecture has led to remarkable advancements in natural language processing. A key advantage of this neural network topology is its ability to retrieve information from any part of the input, thus, constructing rich, contextualized representations. This capability allows the model to effectively deal with long-range dependencies (Tay et al., 2020) and enables in-context learning phenomena, where the model can be adapted to solve downstream tasks using additional input context (Brown et al., 2020; Schick

and Schütze, 2020; Min et al., 2022; Hendel et al., 2023).

Grammatical dependencies, such as subject-verb agreement (Linzen et al., 2016; Warstadt et al., 2020) and coreference resolution (Weischedel et al., 2011), have been extensively used as well-defined tasks to study the contextual abilities of pre-trained language models (Marvin and Linzen, 2018; Tenney et al., 2019b,a; Niu et al., 2022; Kulmizev et al., 2020; Lampinen, 2022). These tasks often require the model to capture and exploit the syntactic relationship between word pairs in the sentence; for example in the case of coreference resolution, the model needs to disambiguate a pronoun with respect to the subject as its *single reference point* in the context. Despite a rich literature on this, the scenario where multiple grammatical cues are present within the context remains underexplored.

In this paper, we use coreference resolution as our case study and analyze model behavior in cases where the context contains multiple sources of information that are relevant for the target task (which we refer to as '*cue*' words), aiming to identify which contextual cues the model prioritizes when disambiguating target pronouns. Consider the following example, in which the last pronoun as a target word that the model is asked to generate is marked in **bold** and all possible cues to disambiguate it ('she' versus 'he') are underlined:

> Mary loves playing the piano. She practices every day, and her music teacher says **she** is very talented.

Specifically, we investigate how the model benefits from various cue words when generating the last target pronoun in the output. To this end, we make use of the Biography corpus as it naturally contains numerous referential expressions that refer to the same individual. using two complementary analytical approaches, we analyze BERT (Devlin et al., 2019) and GPT-2 (Radford et al., 2019), two models with different architectures and training

objectives, across contexts with various numbers of cues, revealing a notable distinction between the behavior of encoder-based and decoder-based language models.

Firstly, we use Value Zeroing (Mohebbi et al., 2023b) as a context-mixing method to track the flow of information from cue words to the representation of the target word at each layer of the model. We find that decoder-based models tend to incorporate the final cue words in the context to form the contextualized representation of the target word. In contrast, encoder-based models rely on the first cue words.

Secondly, we employ a variant of activation patching (Vig et al., 2020a; Geiger et al., 2021b; Meng et al., 2022), a recently popular mechanistic interpretability method (Ferrando et al., 2024; Mohebbi et al., 2024), to measure the impact of each cue word on the model's confidence in generating the target word. While context-mixing methods quantify information mixing in the representation space, the second approach focuses on the language model head to determine whether the encoded information is actually used for prediction.

Our empirical results show that the predictions of the two analysis methods are consistent with each other, implying that the cues that contribute more to the representation of the target word also play a crucial role in the model's final decision. Specifically, our main finding indicates that, in contexts with multiple grammatical cues, encoder-based models tend to prioritize the earlier cues, while decoder-based models rely on the later cue words when disambiguating the target pronoun. [1]

## 2 Related Work

Many analytical studies have been conducted to examine the grammatical capabilities of pre-trained language models, often by probing their layerwise representations for tasks such as part-of-speech tagging (Giulianelli et al., 2018), dependency parsing (Hewitt and Manning, 2019; Chrupała and Alishahi, 2019), subject-verb agreement (Giulianelli et al., 2018), and coreference resolution (Tenney et al., 2019a; Fayyaz et al., 2021). These tasks have also been leveraged in another line of research, particularly as a case study for evaluating attribution methods (Abnar and Zuidema, 2020; Mohebbi et al., 2023b; Ferrando et al., 2022), as they provide a

clear ground truth for assessing the plausibility of attribution scores. For example, when predicting a pronoun, an appropriate attribution method is expected to highlight the subject of the sentence. While these studies focus only on cases with a single plausible cue in the context (e.g., subject) to disambiguate the target word (e.g., pronoun), our work investigates model behavior when multiple sources of information (cue words) exist in the context.

For this purpose, we leverage two state-of-the-art analysis methods from two active lines of interpretability research: one that aims at measuring token-to-token interactions in the model known as *context mixing*, while the other focuses on reverse engineering the model's decision and decompose it to understandable components, known as *mechanistic interpretability*.

**Context mixing.** This line of work focuses on tracking information flow in the model, providing a map score that quantifies token-to-token interactions at each layer. This can be achieved using a group of analytical approaches known as '*context-mixing*' methods. Although self-attention weights are often seen as a straightforward measure of context mixing in Transformers, numerous studies have shown that relying solely on raw attention can be misleading (Bibal et al., 2022; Hassid et al., 2022). They often focus on meaningless and frequently occurring tokens in the input, such as punctuation marks and special separator tokens in models trained on text (Clark et al., 2019), or background pixels in vision Transformers (Bondarenko et al., 2023).[2] Hence, several methods have been developed to broaden the scope of analysis and incorporate other model components into the computation of the context-mixing (Abnar and Zuidema, 2020; Kobayashi et al., 2020, 2021; Ferrando et al., 2022; Modarressi et al., 2022; Mohebbi et al., 2023b).

**Mechanistic interpretability.** This body of research aims to make use of specific characteristics of Transformer architecture and combine them with causal methods to identify specific subnetworks within the model, known as *circuits*, that are responsible for particular tasks (Vig et al., 2020b; Geiger et al., 2021a; Wang et al., 2023; Goldowsky-Dill et al., 2023; Conmy et al., 2023; Heimersheim and Nanda, 2024). In our work, we leverage the concept

---

[1] All code for our data creation and experiments is publicly available at `https://github.com/hamid-amir/CueWords`

[2] See Kobayashi et al. (2020)'s study for an explanation.

of activation patching[3] (Vig et al., 2020a; Geiger et al., 2021b; Meng et al., 2022), a commonly used method from this line of work, which measures the drop in a model's confidence using a contrastive approach. The central idea is to overwrite certain activations in the model during a forward pass with cached activations obtained from another run on the same example with minimal changes (known as a corrupted run) and observe the impact on the model's output. While this method has often been used to identify circuits within the model, we adopt it here to measure token importance for the model's predictions.

## 3 Experimental Setup

In this section, we describe the data and models that we use to set up our experiments.

### 3.1 Data

To measure how the model prioritizes possible cue words within a given context, we need a corpus that includes a diverse range of cue words, each capable of independently disambiguating the target words. We find Biography datasets an ideal case study for this purpose since they naturally describe a single individual, frequently referring to the same subject using referential expressions like pronouns.

We use the test set from the WikiBio (Lebret et al., 2016) dataset[4], which contains biographies extracted from Wikipedia with varying lengths. We clean the dataset by removing HTML tags and automatically annotate the cue words in the context by defining a comprehensive list of gender-specific nouns (e.g., 'actor'/'actress') and gendered pronouns (e.g., 'he'/'she') that can serve as cue words for gender identification. The complete list of potential cue words is presented in Table 1.[5] We categorize our data based on the number of cue words within the context of each example (ranging from 2 to 6), balance the data through undersampling, and then split it into training and test sets. The training set is used solely for fine-tuning the models, while the test set is used for conducting all our experiments. The statistics for the final dataset are provided in Table 3.

---

[3]Other terms have been also used in the literature, including Interchange Interventions, Causal Mediation Analysis, and Causal Tracing.

[4]https://huggingface.co/datasets/michaelauli/wiki_bio

[5]The exclusion of other groups is due to the binary labels in the dataset, rather than a choice by the authors.

| Gender | Words |
|--------|-------|
| Male | he, his, him, himself 
 master, mister, mr, sir, sire, gentleman, lord 
 man, actor, prince, waiter, king 
 father, dad, husband, brother, nephew, boy, uncle, son, grandfather |
| Female | she, her, hers, herself 
 miss, ms, mrs, mistress, madam, ma'am, dame 
 woman, actress, princess, waitress, queen 
 mother, mom, wife, sister, niece, girl, aunt, daughter, grandmother |

Table 1: List of potential cues for gender identification.

### 3.2 Target models

In our experiments, we investigate both encoder-based and decoder-based Transformer (Vaswani et al., 2017) language models. Encoder-based models are trained using masked language modeling, where a certain number of tokens are masked in the input, and the model learns to predict them using bidirectional access to the context. In contrast, decoder-based models are trained autoregressively to predict the next word in the context by conditioning only on the preceding words. This distinction allows us to study how different training objectives influence the way models utilize contextual cues.

We opt for BERT (Devlin et al., 2019) and GPT-2 (Radford et al., 2019) as widely used representative models of each category and analyze them in both pre-trained and fine-tuned setups. For fine-tuning, we employ prompt-based fine-tuning (Schick and Schütze, 2021; Karimi Mahabadi et al., 2022) by calculating the Cross-Entropy loss specifically over the output logits corresponding to a limited set of vocabulary words, particularly male and female pronouns. The accuracy of each model before and after fine-tuning is presented in Table 4.

### 3.3 Model input setup

Consider the following example from the dataset, which includes four cue words marked with underlines:

> Ron Masak is an American actor. He began as a stage performer, and much of **his** work is in theater.

We always ask the model to predict the last pronoun in the context (here, '*his*') as the target word. So, for an encoder-based model, we replace the target word with a special mask token[6]:

> Ron Masak is an American actor. He began as a stage performer, and much of [MASK] work is in theater.

---

[6]The symbol for the masked token depends on the tokenizer used by the model; for BERT, it is [MASK].

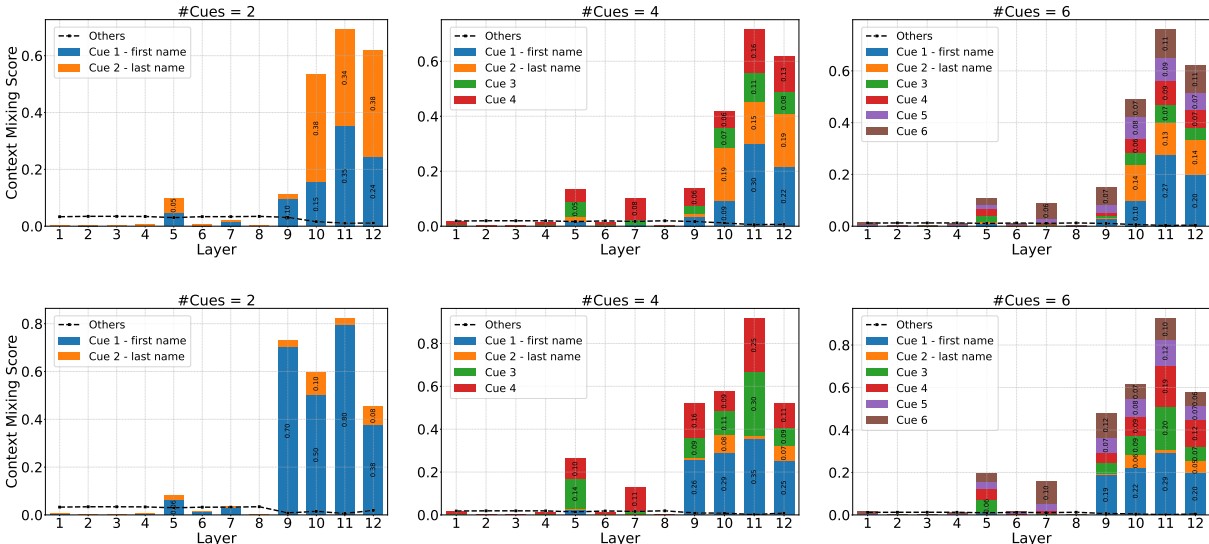

Figure 1: Value Zeroing scores for the **pre-trained** (top row) and **fine-tuned** (bottom row) **BERT** across different numbers of cue words.

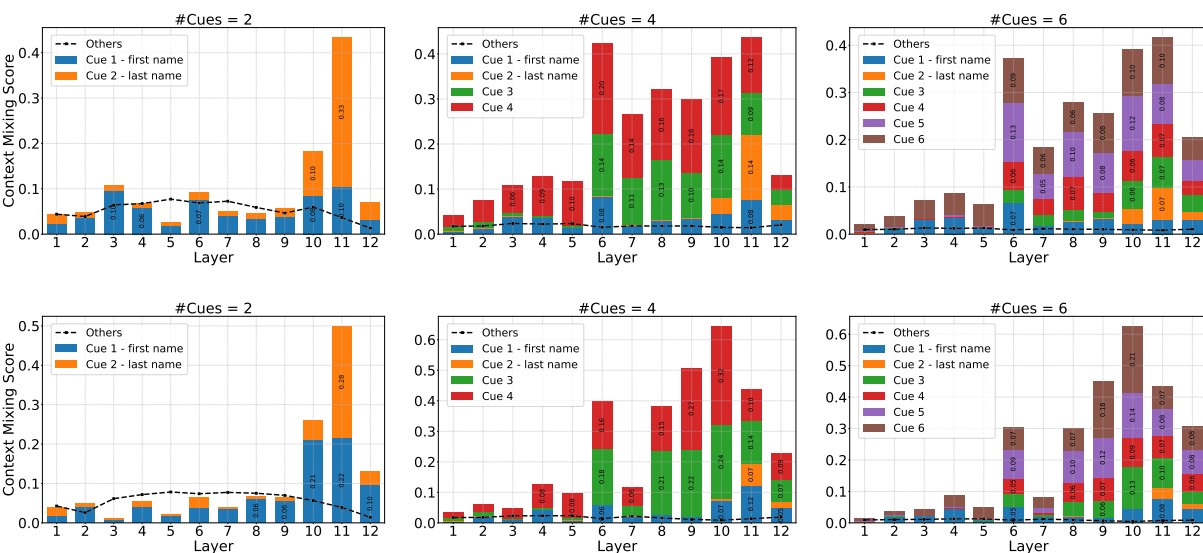

Figure 2: Value Zeroing scores for the **pre-trained** (top row) and **fine-tuned** (bottom row) **GPT-2** across different numbers of cue words.

For a decoder-based model, we keep the sentence up to the last word before the target word and ask the model for the next token prediction:

> Ron Masak is an American actor. He began as a stage performer, and much of

We select those instances where the target word is a pronoun and the model correctly identifies the target word, ensuring accurate gender identification.[7]

In the next sections, we investigate the model internals to understand which contextual cues the model relies on to form representations of the target words and make its final predictions.

## 4 Which cue does the model rely on to form a target representation?

Transformers perform well at retrieving information from any part of the context to build contextualized representations. Our first step is to trace the flow of information within the model to understand how different contextual cues shape the representation of target words.

---

[7]We consider target words to be correct in both their capitalized and lowercase forms.

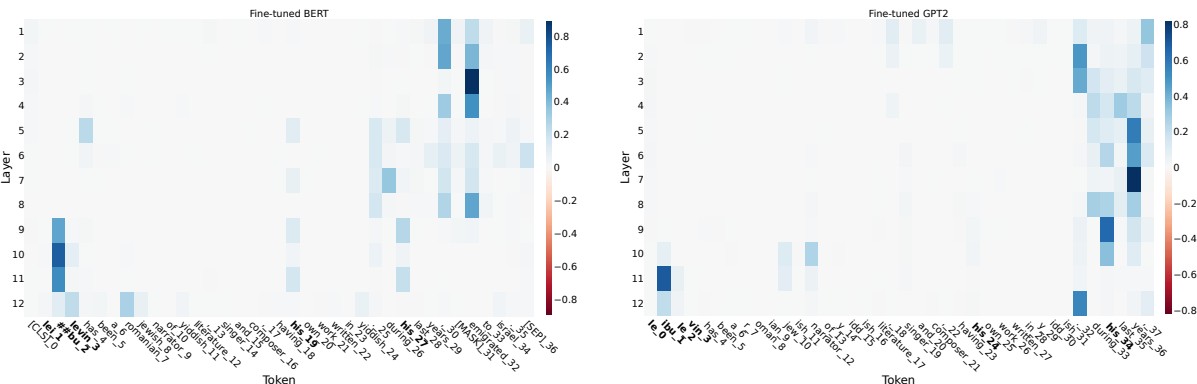

Figure 3: Value Zeroing scores for constructing target token representation in a test example for fine-tuned models. Cue words are highlighted in **bold**.

## 4.1 Setup

We use Value Zeroing (Mohebbi et al., 2023b), a new technique that has shown promise in various domains, including text and speech (Mohebbi et al., 2023a).[8] It iteratively zeroes out the value vector of each token in the context and measures the cosine distance between the modified and original representation of the target word. This distance indicates the degree each token influences the target word's representation—the greater the distance, the higher the contribution.

Using this method, we extract the contribution of each token including the cue word tokens to the representation of the target word at each layer of the model.[9] Scores are normalized to sum to 1 for each context. In encoder-based models, the target position corresponds to the time step of the masked position, while in decoder-based models, it corresponds to the time step when the target token is being generated.

## 4.2 Results

Figures 1 and 2 demonstrate the contribution of cues to the representation of target words in BERT and GPT-2 models, respectively, averaged across all examples in the test set. The analysis covers both their pre-trained (top row) and fine-tuned (bottom row) setups across three scenarios when there are 2, 4, and 6 cues in the context.[10] Additionally,

we report the average context mixing score of non-cue tokens in the context (labeled as 'Others'), as a baseline, to highlight the significance of the cues' contributions.

As shown in Figure 1, BERT significantly incorporates earlier cue words into the representation of the target word, starting from the middle layers. Looking at different scenarios when the number of cues in the context increases, the pre-trained model pays more attention to the first and second cue, while the fine-tuned model pays dominantly to the first cue, compared to other cue words. This suggests that the model tends to keep the first cue in the context as the main source of information for gender identification during the information-mixing process.

We also replicated our experiment by replacing the first two cue words (the first and last names) with their corresponding gendered pronouns ('he' or 'she'). We make this modification to ensure the model's reliance on the first cue word is consistent, regardless of whether the cue is a name or pronoun. The results display the same pattern, confirming our hypothesis; thus, we relegate these findings to Appendix A.4.

The pattern observed in decoder-based models, however, is clearly different. As illustrated in Figure 2, GPT-2 significantly attends to the later cue words in the context, starting from the earlier layers and peaking in the layers closer to the final layer. The later cues make more contributions, while the first cue word has the least contribution, a behavior entirely opposite to that of BERT. Fine-tuning further intensifies this behavior by increasing the importance of the last cue word. Additionally, substituting names with 'he' or 'she' in the experiments does not alter this behavior, indicating that

---

[8]Results for other common methods can be found in Appendix A.3; while, Attention-Norm (Kobayashi et al., 2020) yields results consistent with Value Zeroing, self-attention and Attention Rollout (Abnar and Zuidema, 2020) show a random pattern, confirming the inefficacy of raw attention weights.

[9]If a word is split into multiple tokens by the model's tokenizer, we take the maximum score among them.

[10]Results for other numbers of cues can be found in the Appendix A.3.

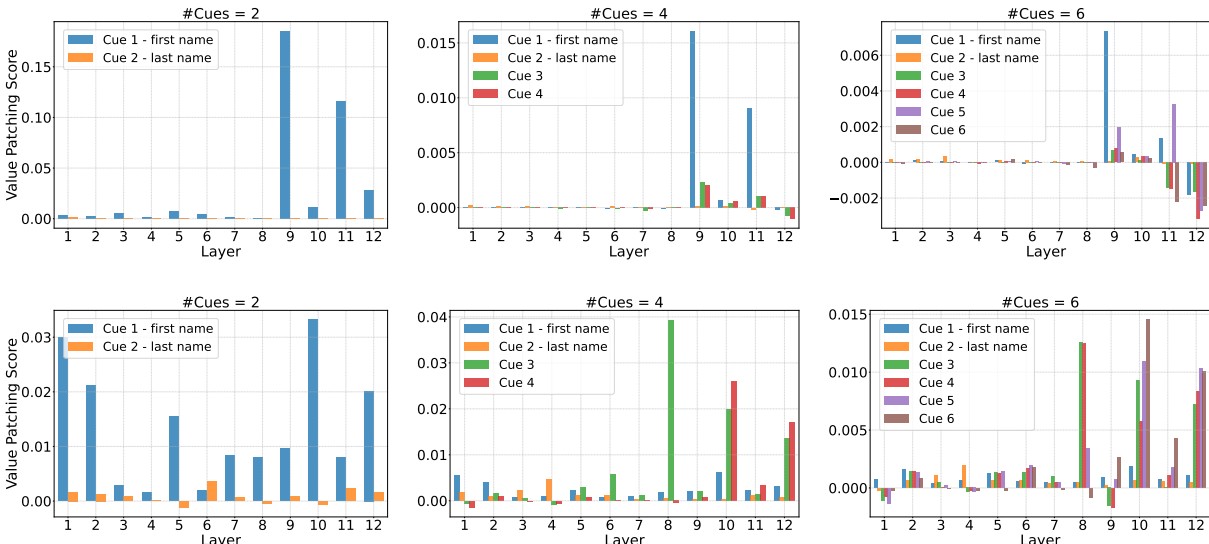

Figure 4: Value patching scores for the **fine-tuned BERT** (top row) and **fine-tuned GPT-2** (bottom row) across different numbers of cue words.

GPT-2 does not exhibit a preference for the first cue word when constructing target token representations, even if it is a pronoun (see Appendix A.4).

In Figure 3, we present the Value Zeroing scores for a test example from the dataset across all layers of both fine-tuned models. The context contains four cues, with BERT primarily using the first cue, which is the first name, to construct the target token representation in the final layers. In contrast, GPT-2 relies on the last cue.[11]

# 5 Which cue does the model rely on to predict a target word?

In Section 4, we quantified context-mixing in the model to assess each cue word's contribution to the contextualized representation of the target word. This analysis reveals how information from these cues is encoded into the target representation. However, it does not show whether this information is actually used during inference when predicting the target token. The prediction process (masked or next token prediction) in the model is typically performed by a trained language model head which takes the target representation and generates logits for all tokens in the vocabulary. The goal here, in our second step, is to involve the model's prediction in the analysis to investigate how different contextual cues influence the model's decision.

## 5.1 Setup

Activation patching can be applied to various components of a model, including attention heads, MLP outputs, and residual streams. In this study, however, the focus is on patching value vectors within a Transformer layer. The reason for this choice is to keep the pattern of attention (and thus the flow of information) in the model intact, and only nullify the value of a specific cue token representation in a given context. Replacing a token from a clean run with one from a corrupted run adds confounding variables, as it introduces a different pattern of attention that may not match those of the clean run.

We treat the original text in the dataset as clean text and generate corrupted text by replacing all cue words in the clean text with their gender-opposite counterparts. For each cue word, a corresponding counterpart exists, as shown in Table 1, except for the first and second cue words, which are first and last names, respectively. In these cases, we substitute the names with a constant name with the opposite gender, ensuring the same number of subwords in all of our model's tokenizers (see Table 2). An example of clean and corrupted text from our dataset is shown below:

> clean:
> Ron Masak is an American actor. He began as a stage performer, and much of **his** work is in theater.

> corrupted:
> Amy Willinsky is an American actress. She began as a stage performer, and much of **her** work is in theater.

---

[11]In this particular example, GPT-2 also utilizes the first cue, highlighting the variance in the results.

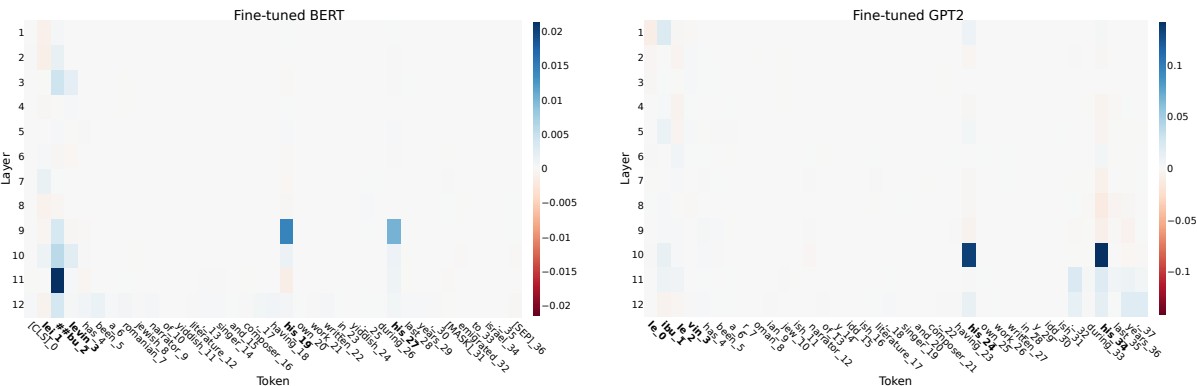

Figure 5: Value patching scores for a test example for fine-tuned models. Cue words are highlighted in **bold**.

| Name type | Gender | #Tokens | Constant name |
|-----------|--------|---------|---------------|
| First name | Male | 1 | Bob |
| | | 2 | John |
| | Female | 1 | Amy |
| | | 2 | Noora |
| Last name | - | 1 | Walker |
| | | 2 | Willinsky |

Table 2: A set of random constant names based on gender and the number of tokens into which the word is split.

We generate corrupted texts for each example in the test dataset, input them into the model, and cache the resulting value vectors for each token as "*corrupted value vectors*." Subsequently, we input the clean text into the model and record the output probability for the target token ($p_t$). We then input the clean text again, but this time, we replace the value vector of a specific token at the time step $j$ at a particular layer with its corresponding corrupted value vector and measure the resulting output probability for the target token ($p_t^{\neg j}$). This process is repeated for all tokens across all layers to measure the value patching score: $p_t - p_t^{\neg j}$. Intuitively, if a cue token is important for the model's prediction, replacing its value vector with a corrupted one (which implies an opposite gender) would lead to a drop in the model's confidence in identifying the true gender.

### 5.2 Results

Figure 4 shows the layer-wise value patching scores of the cue words for fined-tuned BERT and GPT-2 across three scenarios when there are 2, 4, and 6 cues in the context.[12] The scores are averaged over

---

[12]Results for pre-trained models and also other number of cues can be found in Appendix A.5.

all examples in the test set.

BERT exhibits a significant loss of confidence in generating the correct target word when the value vector of the first cue word is replaced with that from a corrupted run, compared to the other cue words.

In contrast, GPT-2 exhibits an opposite pattern, with later cues in the context playing a more influential role in the model's decision-making. There is one exception for GPT-2: when only two cue words are present in the context, patching the first cue word affects the model's predictions more than patching the second. This may be reasonable for a decoder-based model that sees only prior words, as the last name of a person is not an indicator of gender unless the model has memorized it during pre-training.

In Figure 5, we present the value patching scores for a test example from the dataset across all layers of both models. There are four cues present in the context, all of which change the model's confidence when their value vectors are patched. Yet, we can see the second sub-word of the first cue is particularly significant for BERT, while the final cue word is the major player for GPT-2 in making their respective decisions.

## 6  Conclusion

In this paper, we examined how language models handle gender agreement when multiple valid gender cue words are present in the context. We carried out extensive experiments using two state-of-the-art and complementary analytical approaches on two prominent language models with different model architectures: BERT and GPT-2. Our results suggest that encoder-based and decoder-based models behave differently in prioritizing contextual cues. More specifically, we observed that BERT

mainly relies on the earlier cues in the context, while GPT-2 mostly uses the later ones. These findings can be explored and leveraged in future to enhance model efficiency (by excluding redundant cues from the computations), update the models' beliefs (by intervening with the most crucial cues they rely on), or improve the way we interact with them through prompting (by considering the impact different cues may have in various positions within a given context).

## 7  Limitations

Our experiments and findings are drawn based on a grammatical agreement task as a well-defined scenario where multiple cues exist in a context. This choice was made because it allows us to identify and annotate cue words using NLP tools automatically. Alternatively, other case studies, such as Question Answering datasets, where multiple cues in the context refer to the answer could be explored in future work.

Furthermore, we ran our experiments on two widely used language models but with base size (due to our limited computational budget). Future work can extend these experiments to include more recent, large language models as well.

## Acknowledgements

We gratefully acknowledge the Speech and Language Processing Lab, led by Dr. Hossein Sameti at Sharif University of Technology for providing the computational resources used in running the experiments. Hamidreza Amirzadeh extends his special thanks to Mr. Shirzady for his assistance with the GPU server. Hosein Mohebbi is supported by the Netherlands Organization for Scientific Research (NWO), through the NWA-ORC grant NWA.1292.19.399 for '*InDeep*'.

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

# A Appendix

| #Cues | Train Set | Test Set |
|-------|-----------|----------|
| 2 | 2480 | 1677 |
| 3 | 1439 | 934 |
| 4 | 921 | 629 |
| 5 | 638 | 438 |
| 6 | 505 | 287 |

Table 3: Distribution of training and test examples across different numbers of cues before downsampling

| Model | Accuracy | |
|-------|----------|----------|
| | Pre-trained | Fine-tuned |
| BERT | 86.6 | 97.8 |
| GPT-2 | 66.7 | 77.9 |

Table 4: The accuracy of pre-trained and fine-tuned models on our test set

## A.1 Dataset Statistics

Table 3 presents the distribution of examples for each cue word in our dataset. To ensure balanced representation, we downsampled the examples for cue words 2 through 5 so that each category has an equal number of instances as those with 6 cues. Consequently, our final training set includes 505 examples per cue word, yielding a total of 2525 train examples. Similarly, the test set comprises 287 examples per cue word, resulting in a total of 1435 test examples.

## A.2 Models Accuracy

Table 4 shows the accuracy of pre-trained and fine-tuned models on our test set. BERT outperforms decoder-based model GPT-2 mainly because it has access to the full context of each example, including tokens that follow the target word. In contrast, decoder-based models lack this advantage.

## A.3 Context Mixing Scores

In Figures 6 to 19, we present the context mixing scores derived from various methods used in our study, including self-attention weights, Attention Rollout, Attention Norm, and Value Zeroing. These results are displayed for all different number of cue words and all the models we analyzed.

Please note that there is currently no implemented version of Attention Norm for decoder-based models, so we were unable to provide Attention Norm results for GPT-2.

## A.4 Context Mixing Scores: Ablation Study

In our primary experiments, we observed that BERT predominantly utilizes the first name as the main contributor to constructing mask token representations. To determine whether this significance is due to the first cue position or the specific use of a first name, we conducted an ablation study. In this study, we removed the last name and replaced the first name with "he" or "she," depending on the gender of the example. Figures 20 and 21 display the context mixing scores from this ablation study for both the pre-trained and fine-tuned BERT models. As these figures indicate, there is no significant shift towards the last cues, leading us to conclude that the importance lies in the cue being first, rather than it being a first name. Additionally, we conducted this experiment with GPT-2 as well, and once again, the results showed no significant difference compared to original experiments (see Figures 22 and 23). This suggests that GPT-2 does not depend on the first cue words (not necessarily first names) for constructing target token representations.

## A.5 Value Patching Scores

In Figures 24 to 27, we provide value patching scores for all different number of cue words and models we examined.

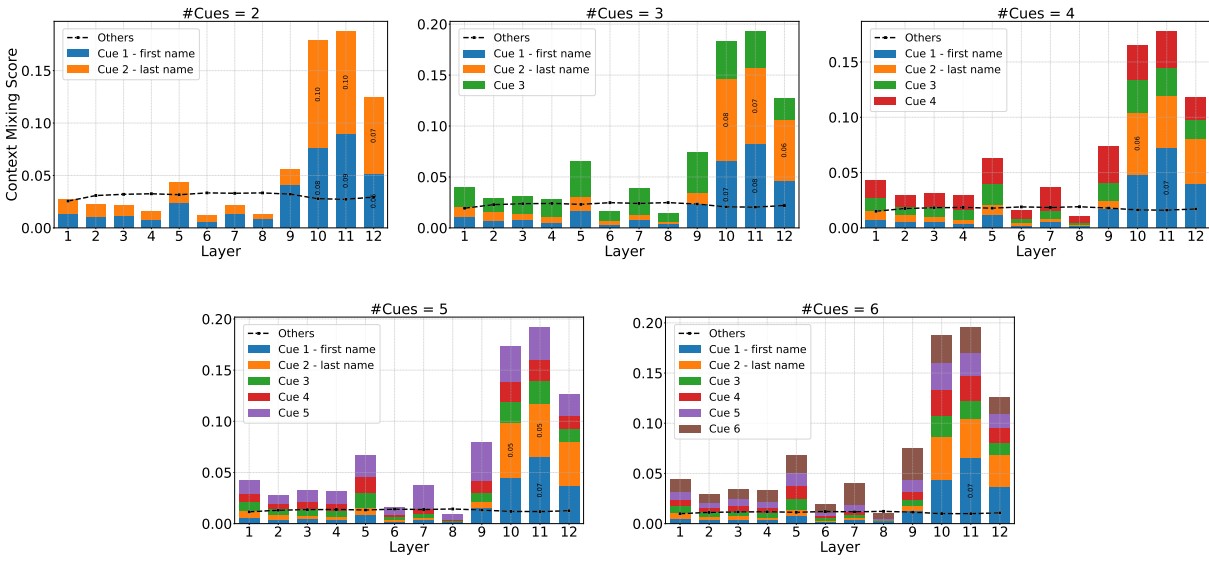

Figure 6: **Self-attention weights** context mixing scores for the **pre-trained BERT** model across different numbers of cue words

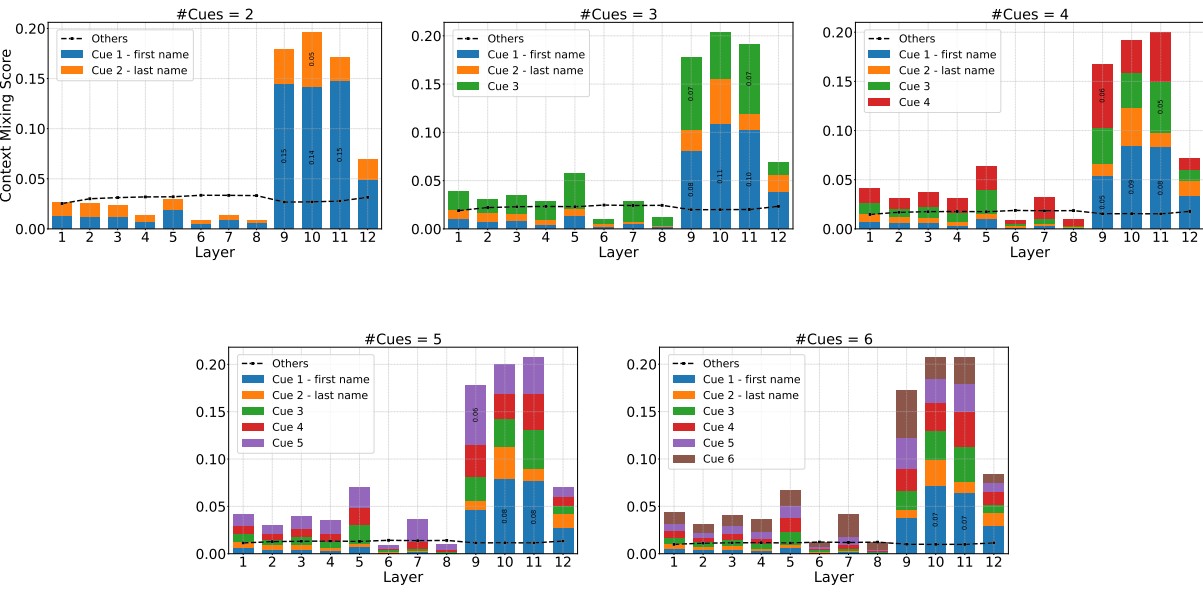

Figure 7: **Self-attention weights** context mixing scores for the **fine-tuned BERT** model across different numbers of cue words

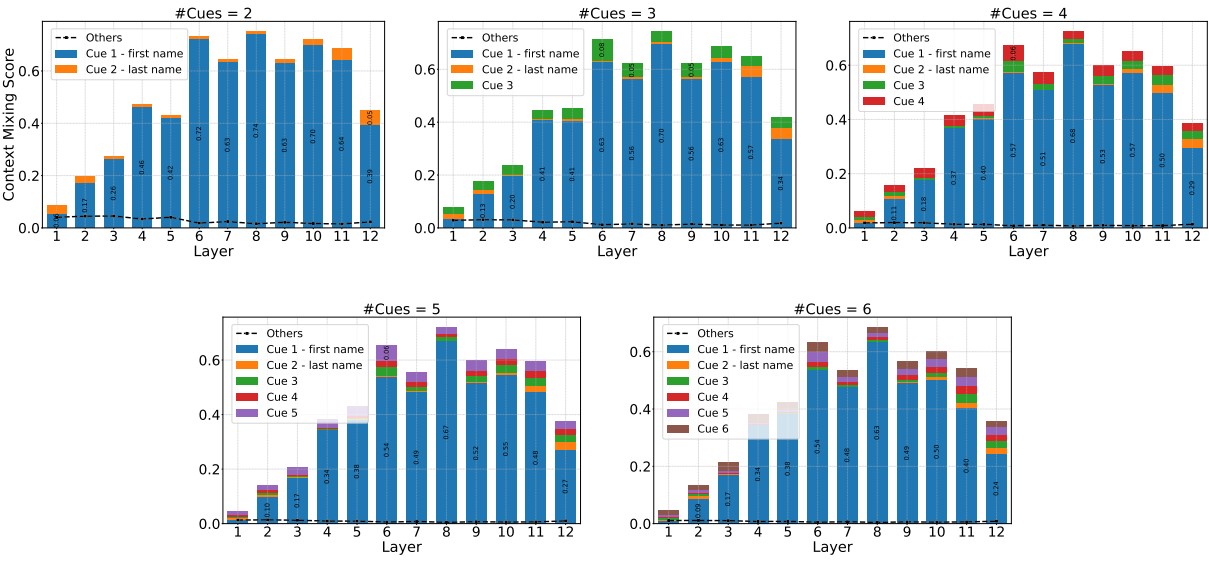

Figure 8: **Self-attention weights** context mixing scores for the **pre-trained GPT-2** model across different numbers of cue words

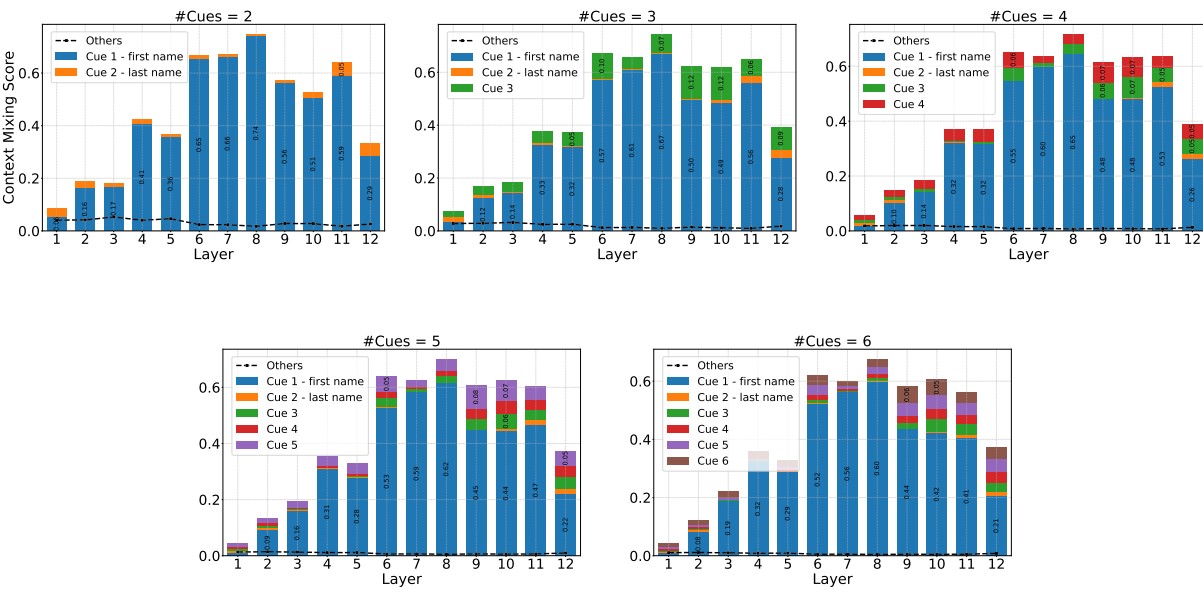

Figure 9: **Self-attention weights** context mixing scores for the **fine-tuned GPT-2** model across different numbers of cue words

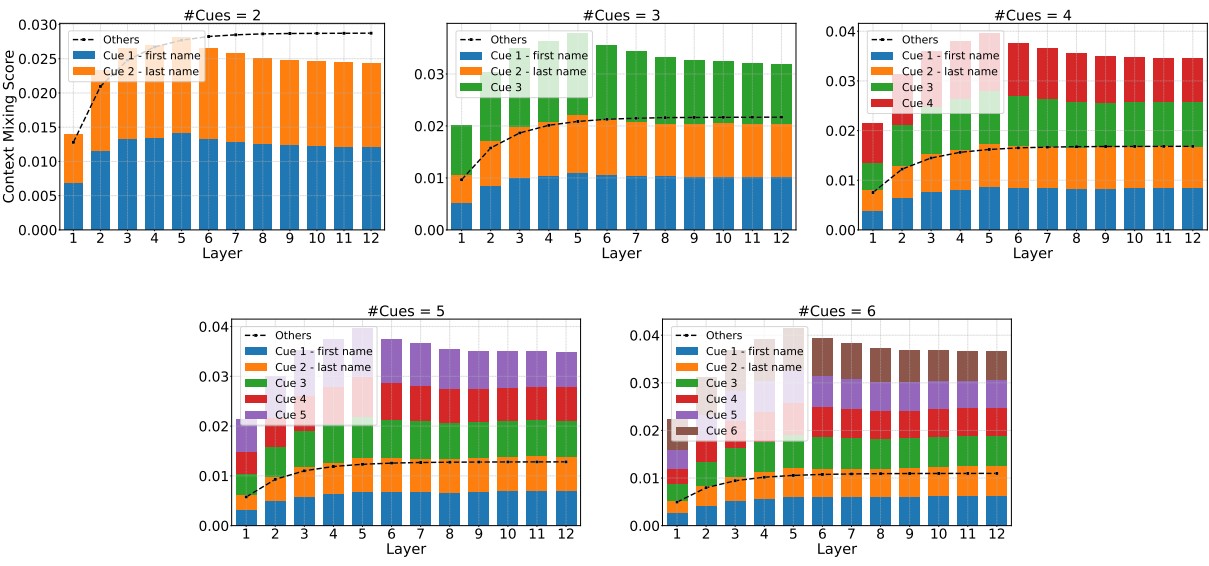

Figure 10: **Attention Rollout** context mixing scores for the **pre-trained BERT** model across different numbers of cue words

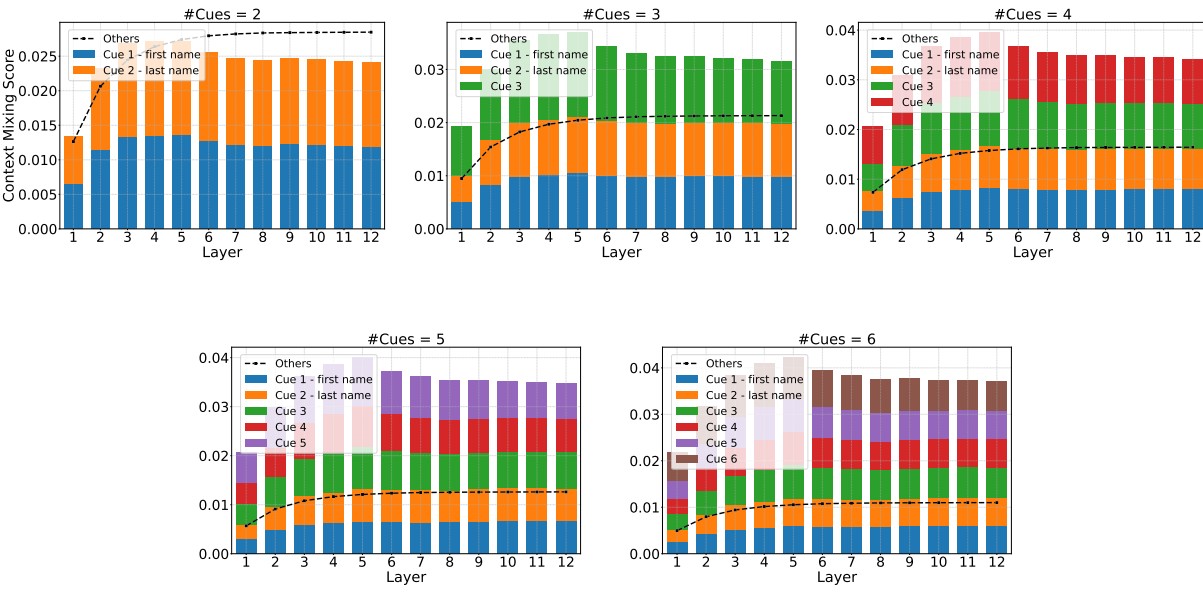

Figure 11: **Attention Rollout** context mixing scores for the **fine-tuned BERT** model across different numbers of cue words

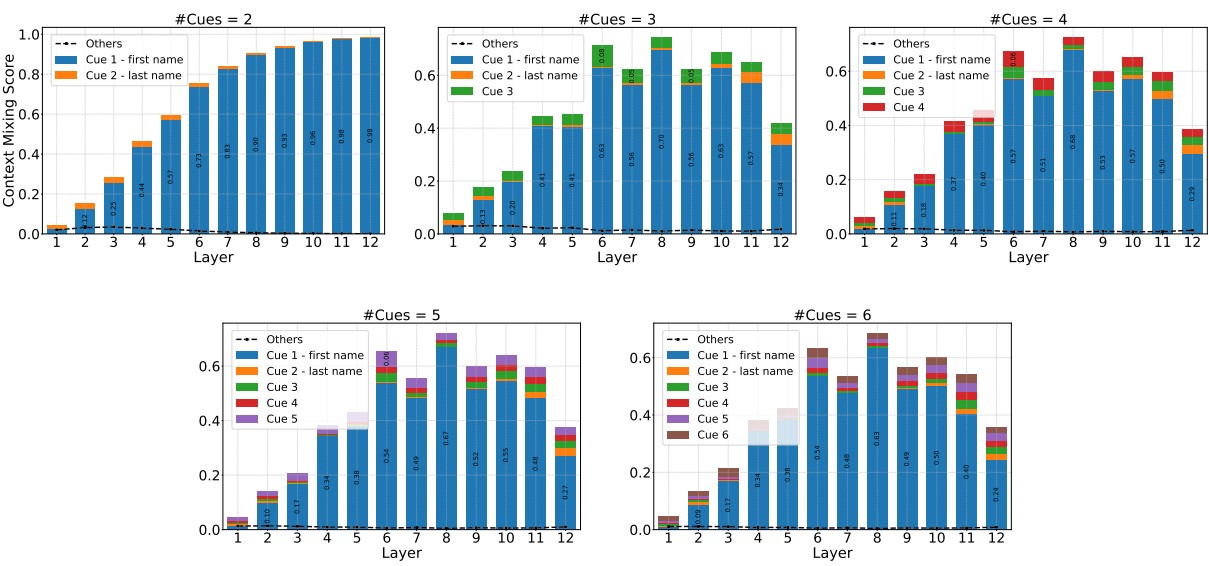

Figure 12: **Attention Rollout** context mixing scores for the **pre-trained GPT-2** model across different numbers of cue words

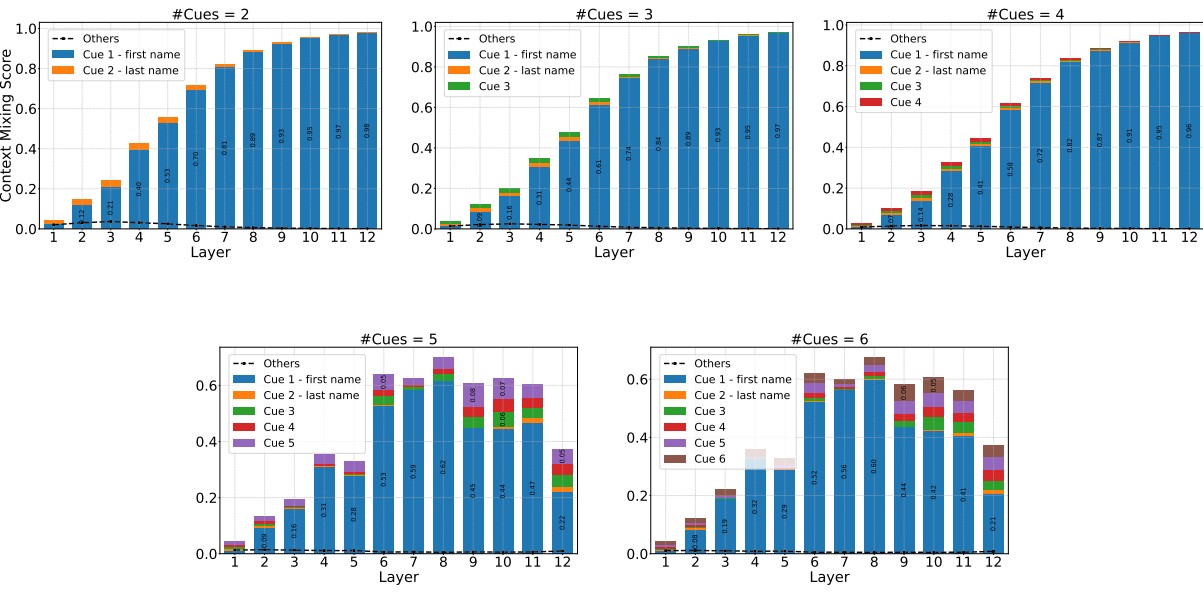

Figure 13: **Attention Rollout** context mixing scores for the **fine-tuned GPT-2** model across different numbers of cue words

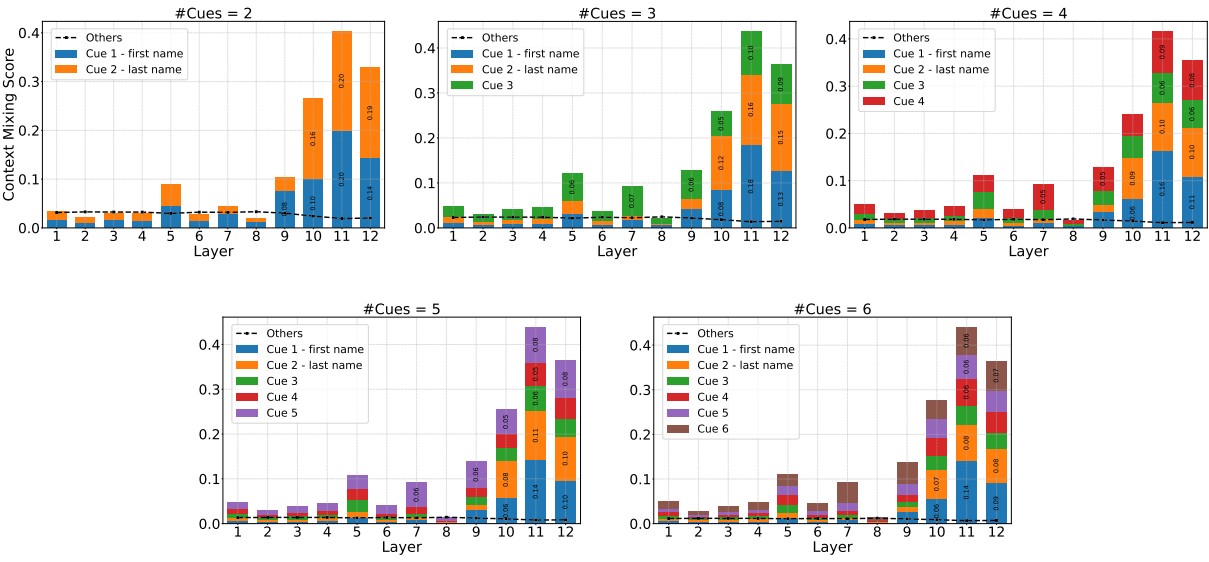

Figure 14: **Attention Norm** context mixing scores for the **pre-trained BERT** model across different numbers of cue words

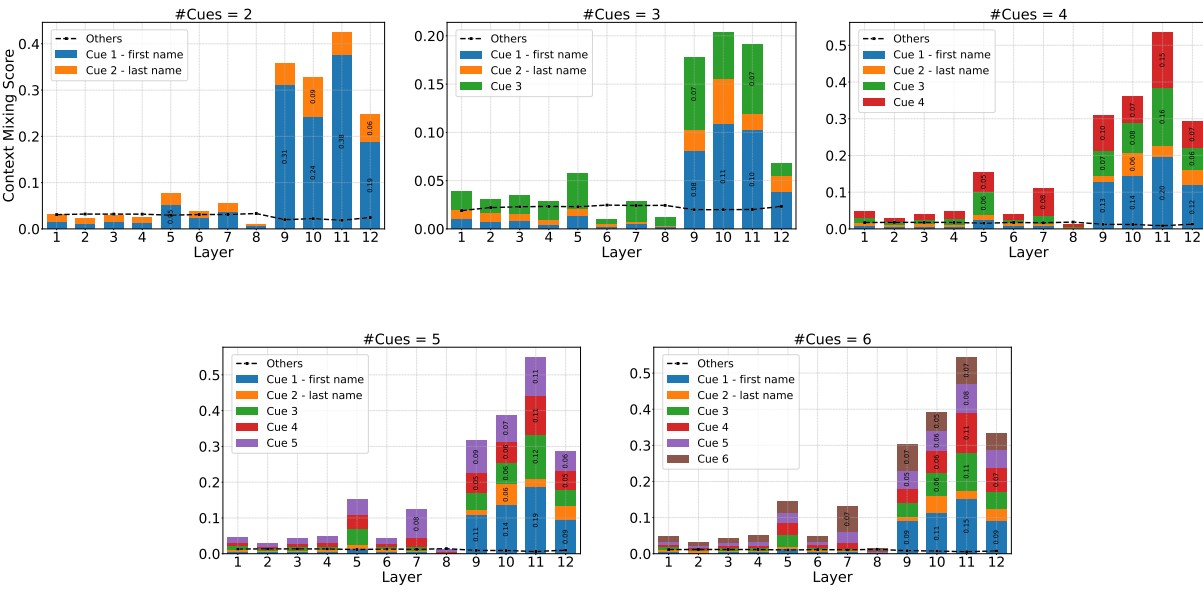

Figure 15: **Attention Norm** context mixing scores for the **fine-tuned BERT** model across different numbers of cue words

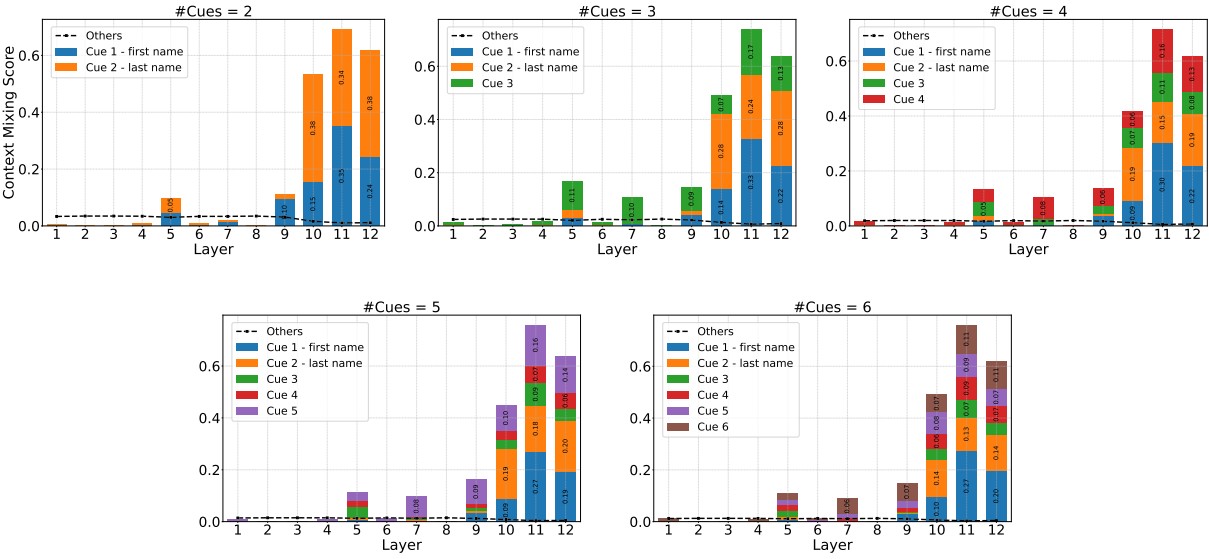

Figure 16: **Value Zeroing** context mixing scores for the **pre-trained BERT** model across different numbers of cue words

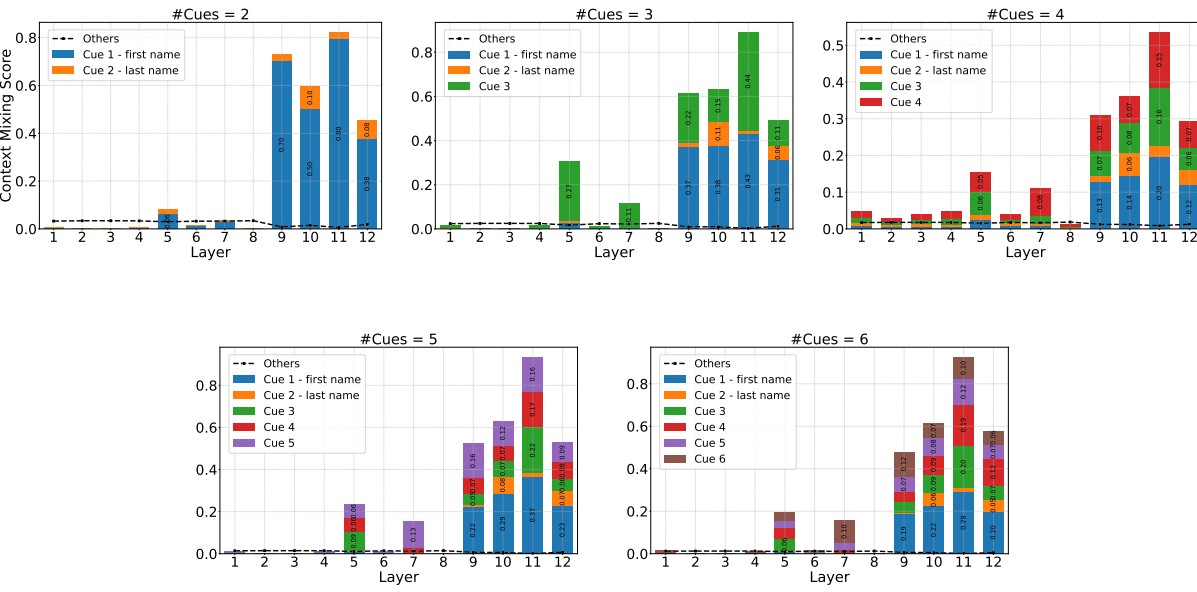

Figure 17: **Value Zeroing** context mixing scores for the **fine-tuned BERT** model across different numbers of cue words

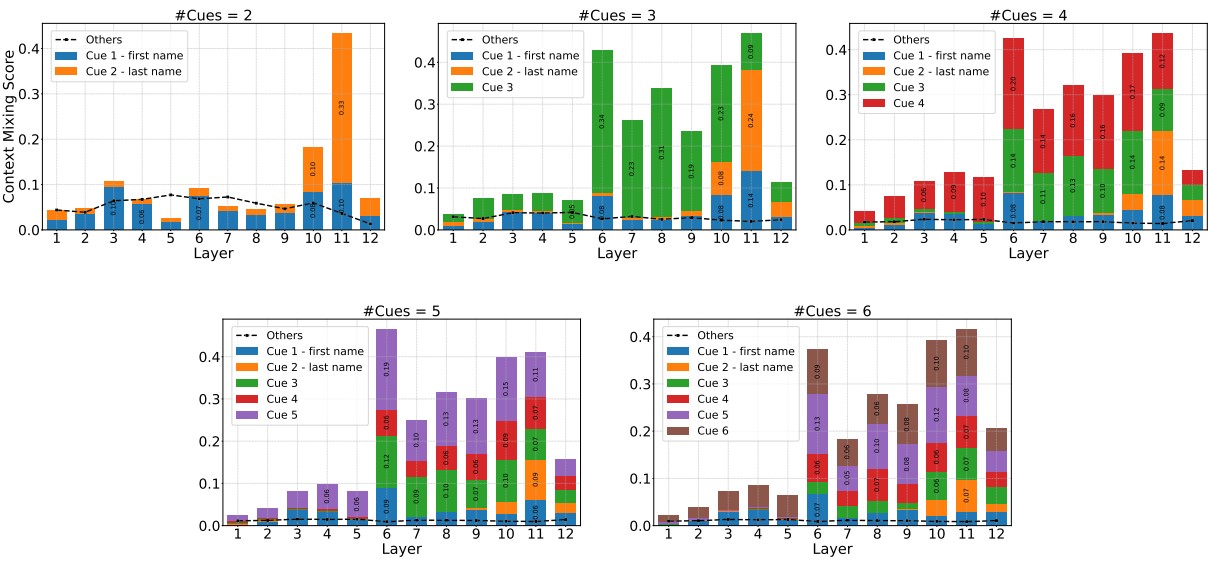

Figure 18: **Value Zeroing** context mixing scores for the **pre-trained GPT-2** model across different numbers of cue words

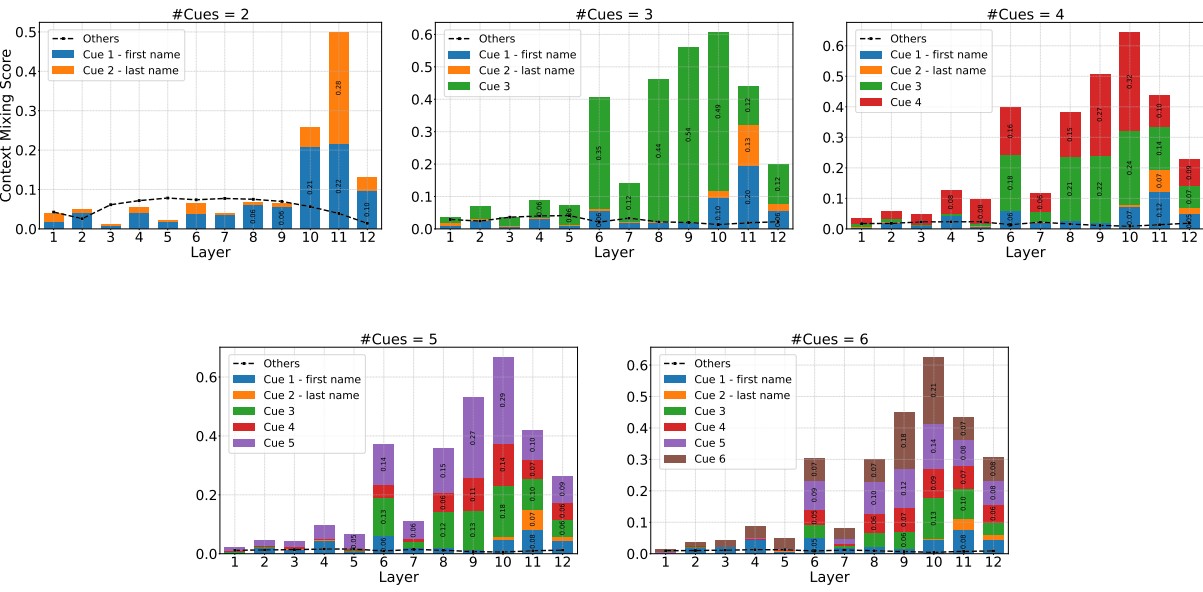

Figure 19: **Value Zeroing** context mixing scores for the **fine-tuned GPT-2** model across different numbers of cue words

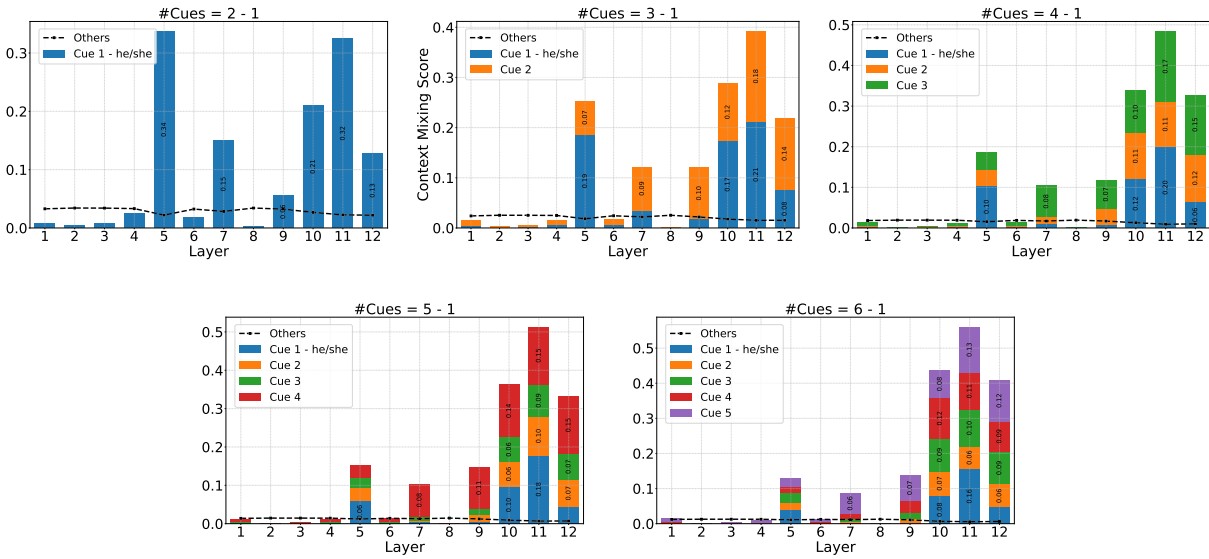

Figure 20: **Value Zeroing** context mixing scores for the **pre-trained BERT** model with varying cue word counts, when removing the last names and **replacing first names with "he/she."** Note: Removing the last name results in the loss of a cue word.

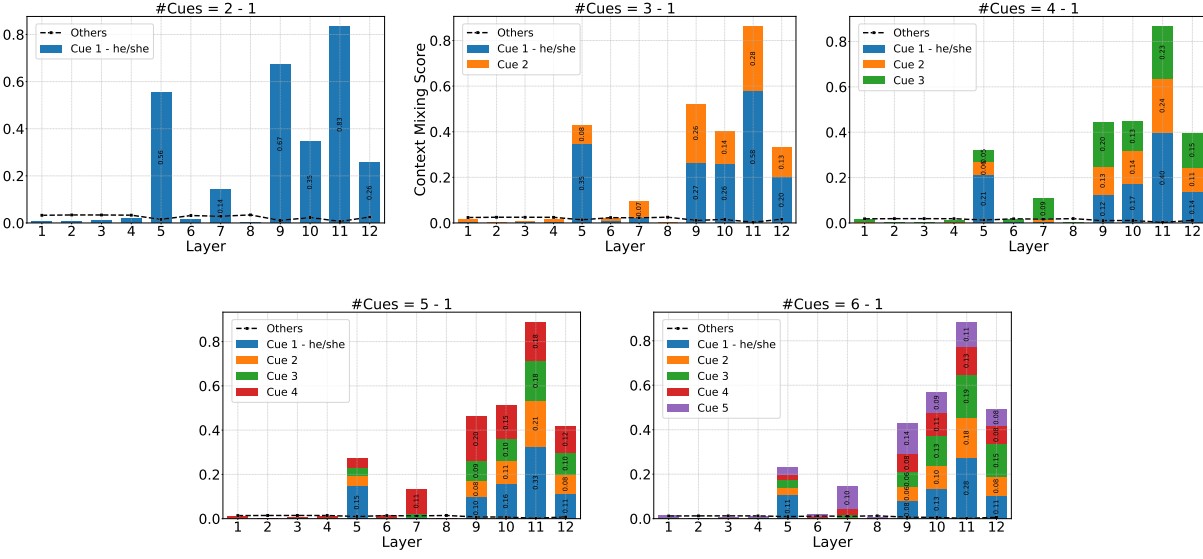

Figure 21: **Value Zeroing** context mixing scores for the **fine-tuned BERT** model with varying cue word counts, when removing the last names and **replacing first names with "he/she."** Note: Removing the last name results in the loss of a cue word.

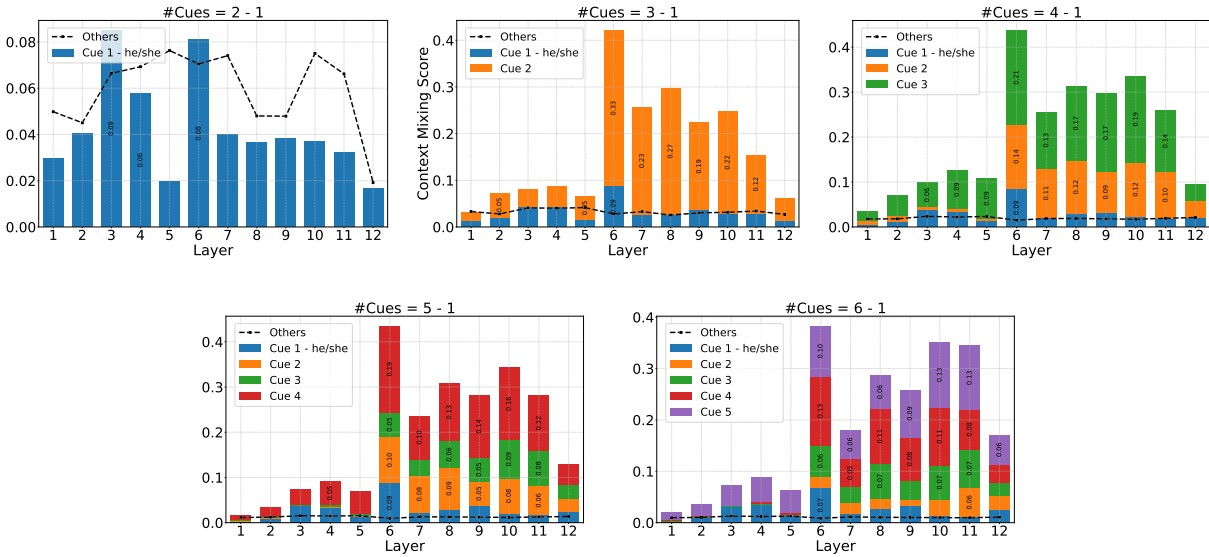

Figure 22: **Value Zeroing** context mixing scores for the **pre-trained GPT-2** model with varying cue word counts, when removing the last names and **replacing first names with "he/she."** Note: Removing the last name results in the loss of a cue word.

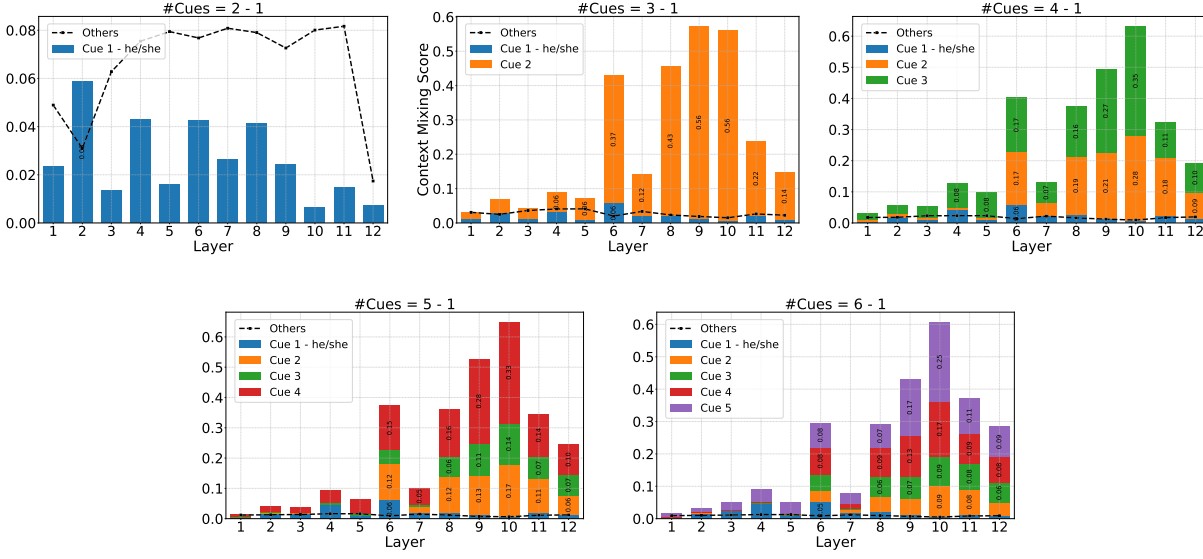

Figure 23: **Value Zeroing** context mixing scores for the **fine-tuned GPT-2** model with varying cue word counts, when removing the last names and **replacing first names with "he/she."** Note: Removing the last name results in the loss of a cue word.

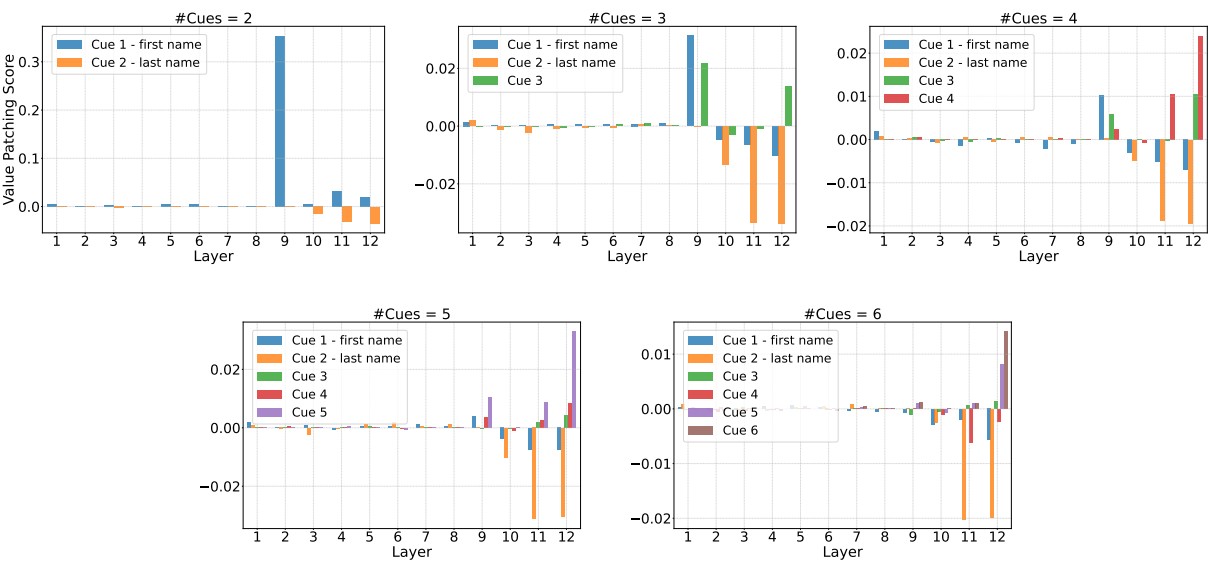

Figure 24: **Value patching** scores for the **pre-trained BERT** model across different numbers of cue words

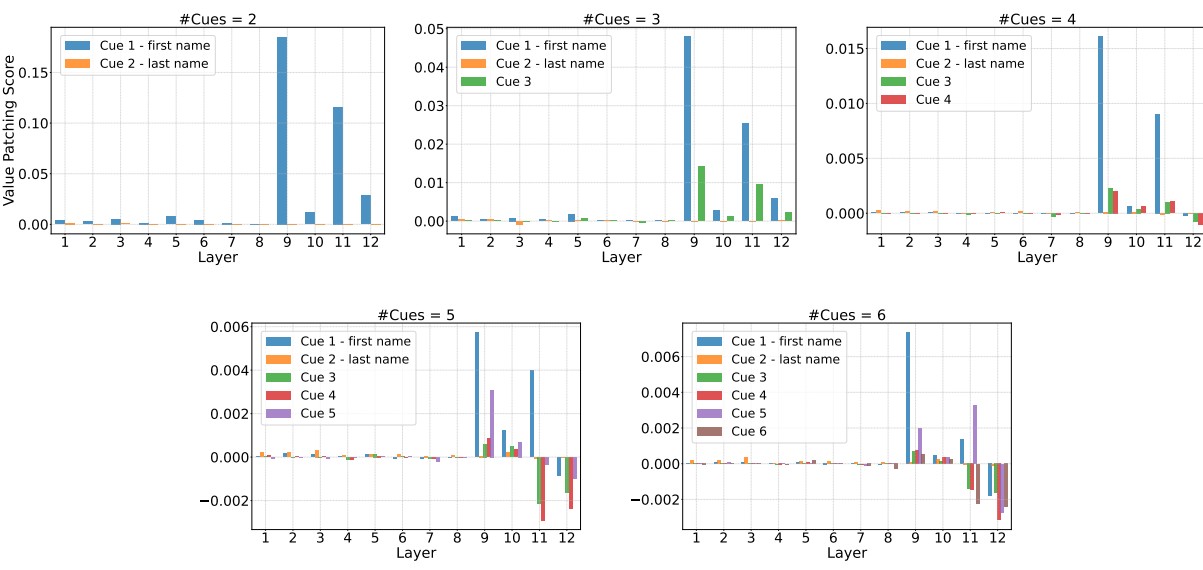

Figure 25: **Value patching** scores for the **fine-tuned BERT** model across different numbers of cue words

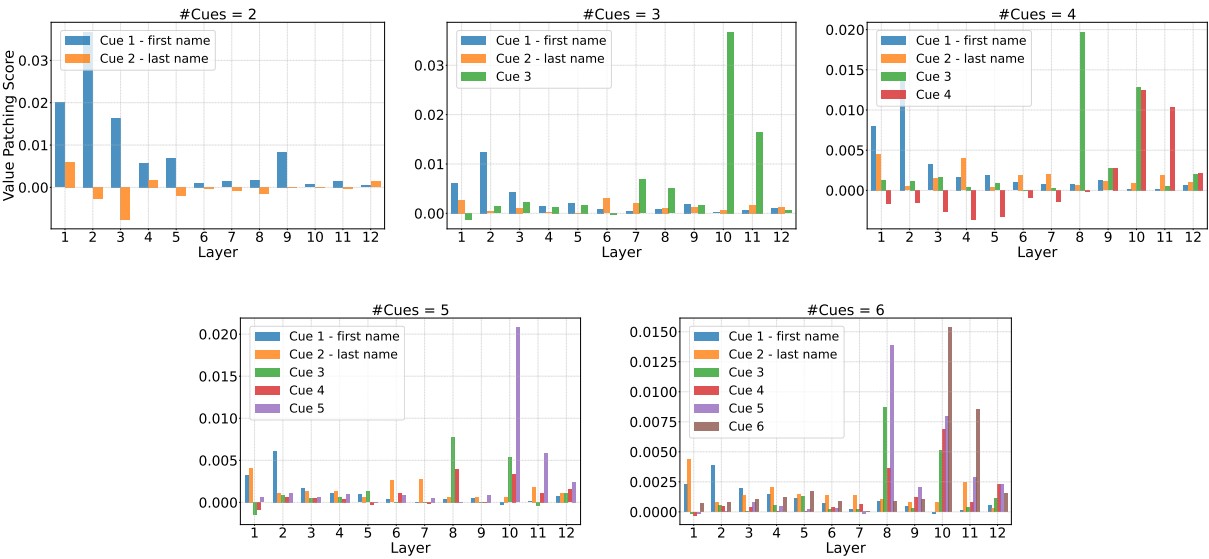

Figure 26: **Value patching** scores for the **pre-trained GPT-2** model across different numbers of cue words

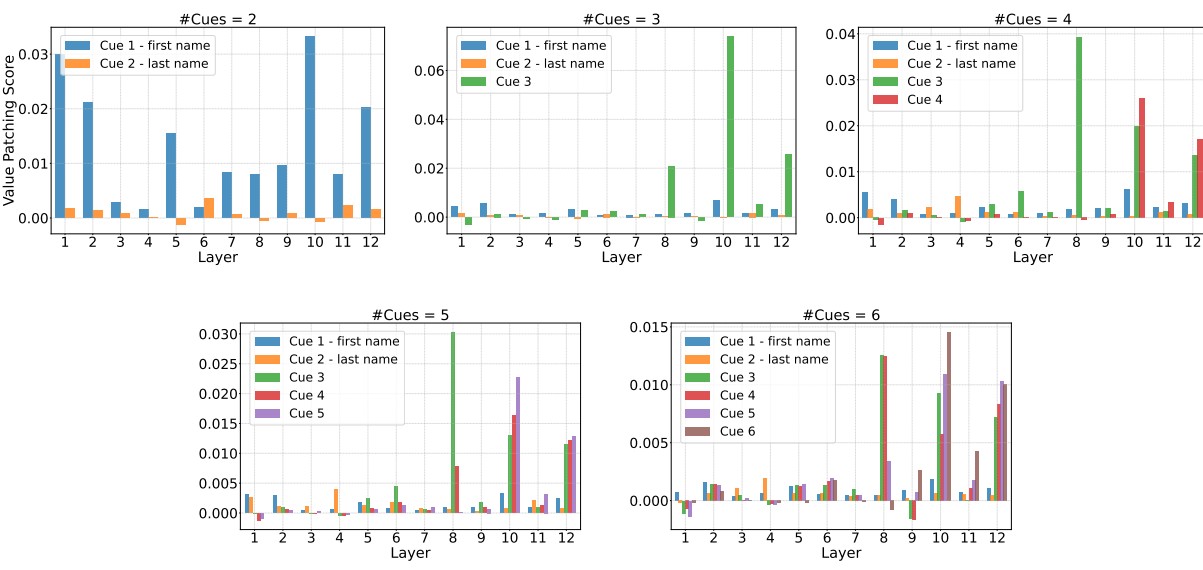

Figure 27: **Value patching** scores for the **fine-tuned GPT-2** model across different numbers of cue words