# OpenReview forum: "How Language Models Prioritize Contextual Grammatical Cues?"
_EMNLP/2024/Workshop/BlackBoxNLP — BlackboxNLP 2024_

### Official Review · Reviewer_bjRQ · 2024-09-07

**Overall Assessment:** 4
**Confidence:** 3

**Best Paper:**

1

**Best Paper Justification:**

N/A

**Comments Questions Suggestions And Typos:**

N/A

**Paper Summary:**

The paper studies how encoder-based and decoder-based language models handle gender agreement when multiple gender cue words appear in the context. By using two popular interpretability techniques: context-mixing via Value-Zeroing and activation patching, the authors analyze how much information about the cues are present in the representations and how much they influence the output. Using a dataset of biographies, the results show that encoder-based LMs (BERT) rely on earlier cues in the context, while decoder-based LMs (GPT-2) make use of cues closer to the final prediction.

**Summary Of Strengths:**

- The paper is well-written and easy to follow.
- The experimental methodology is rigorous and well-designed, and effectively demonstrates how different architectures handle the presence of cues for gender agreement.
- The use of two complementary interpretability tools- context-mixing and an activation patching-, and the fact the conclusions match, strengthens the study's conclusions and provides a more comprehensive understanding of the underlying mechanisms.

**Summary Of Weaknesses:**

- While the study offers valuable insights, the practical implications of these findings remain unclear. The paper would benefit from a more detailed motivation on why studying this task is relevant.
- I missed some discussion about why different architectures might rely on cues in a different manner.

---

### Official Review · Reviewer_ZESk · 2024-09-08

**Overall Assessment:** 4
**Confidence:** 4

**Best Paper:**

2

**Best Paper Justification:**

It's an extension of the typical kind of context-sensitivity study to a multiple-clue case; we need to explore more of that!

**Comments Questions Suggestions And Typos:**

None

**Paper Summary:**

The paper discusses context-reliance of two models -- BERT and GPT-2. It focusses on pronoun gender choice in cases where context contains multiple clues for the referent gender. The question is then, which of the multiple clues the model more heavily relies on, and whether there are systematic differences between models in this respect. To investigate that, the authors use two interpretability techniques -- value zeroing and activation patching. The two methods agree in that BERT relies more heavily on early clues, while GPT-2 relies on later ones.

**Summary Of Strengths:**

This is an interesting and carefully conducted series of experiments, with insightful results. The choice of methods is well motivated; the dataset seems adequate for the goal; the results are shown layer-wise and both for pre-trained and fine-tuned versions of the models, which makes it easier to grasp the intuition about how pronominal gender information is treated in the models. The paper is written really well and is very clear.

**Summary Of Weaknesses:**

I don't have a lot of complaints, apart from the obvious limitation that the authors themselves also note -- conclusions are drawn about how different types of models (encoders vs decoders) treat multiple clues, but we don't really know if this observation generalizes beyond these two particular models. It'd be great to have at least two more models to be able to say this (one more encoder, one more decoder). Maybe instead of comparison between pre-trained and fine-tuned versions (although, maybe not).

What I find a bit lacking is a discussion of wider implications of this finding. Encoders rely more on first clues, decoders more on most recent clues -- so what? (Not asking this in a bad way!) What other behaviours do we expect this to connect to?

Finally, a small thing: What about cases where there are two referents of the same gender in a text? Do you have those? Do you try to filter them out? Does it matter (probably not or not much)

---

### Official Review · Reviewer_n6SC · 2024-09-10

**Overall Assessment:** 4
**Confidence:** 3

**Best Paper:**

1

**Best Paper Justification:**

N/A

**Comments Questions Suggestions And Typos:**

N/A

**Paper Summary:**

The work addresses the specific problem of grammatical gender agreement across referents in language model predictions. Its motivation is to extend past studies to a setting with multiple referents to determine how information from subsequent referents affects the model's output?
To address this question, the authors utilize two established interpretability tools: value zeroing and value patching. The first allows tracking the information flow within the model, while the second, thanks to counterfactual analysis, allows pinpointing the referents most important to the model's output.

Both methods show that a masked language model (BERT) relies more on earlier referents, while a causal language (GPT-2) relies on later ones to disambiguate the gender in coreference.

**Summary Of Strengths:**

- The analysis is done only on the relatively old and small instances of language models. It would be interesting to see if the observations hold also for the recent LLMs
- Information flows in LMs should be studied in the context of accuracy to determine whether such information is sufficient to predict the correct gender (e.g., by comparing the results of context mixing with accuracy).

**Summary Of Weaknesses:**

- Clear cut research problem with adequate set of tools and experiment with informative results.
- The methodology and experimental settings are described clearly and in-detail
- The paper analyze multiple distinct models: masked LM (Bert), casual LM (GPT-2) with and without fine-tuning

---

### Decision · Program_Chairs · 2024-09-20

**Decision:**

Accept

**Comment:**

All reviewers agree that this is a solid contribution to BlackboxNLP. Reviewers ZESk and bjRQ both would have liked to have seen a broader reflection on the implication of the authors' findings, and I encourage the authors to take this concern into account in a revised version of either the introduction or conclusion of the paper.